# Neurological Damage Measured by S-100b and Neuron-Specific Enolase in Patients Treated with Electroconvulsive Therapy

**DOI:** 10.3390/brainsci14080822

**Published:** 2024-08-16

**Authors:** Ángel A. Ruiz-Chow, Carlos J. López-Cruz, Daniel Crail-Meléndez, Jesús Ramírez-Bermúdez, José Santos-Zambrano, Laura A. Luz-Escamilla

**Affiliations:** Departamento de Psiquiatría, Instituto Nacional de Neurología y Neurocirugía “Manuel Velasco Suárez”, Insurgentes Sur 3877, Col. La Fama, Ciudad de México C.P. 14269, Mexico; aaruizchow@gmail.com (Á.A.R.-C.); clopez@innn.edu.mx (C.J.L.-C.); jesusramirezb@yahoo.com.mx (J.R.-B.);

**Keywords:** electroconvulsive therapy, neuron-specific enolase, S-100b protein, neuronal damage

## Abstract

Electroconvulsive therapy (ECT) is considered one of the most effective treatments for psychiatric disorders. ECT has proven effective in the treatment of depression, mania, catatonia and psychosis. It is presumed that seizures induced during ECT administration cause toxicity and potentially neuronal and glial cell death. A broad range of neurological disorders increase cerebrospinal fluid and serum levels of neuron-specific enolase (NSE) and S-100b protein. This study aims to investigate the effect of ECT on NSE and S-100b levels, which, together, serve as a proxy for neuronal cell damage. Serum concentrations of S-100b and NSE of adult patients who received ECT were measured by immunoluminometric analysis before and after treatment. A two-way ANOVA test was used to estimate the statistical differences in marker concentrations between the subgroups of the study population. Results: A total of 55 patients were included in the analysis: 52.73% (n = 29) were diagnosed with depression, 21.82% (n = 12) with schizophrenia or other psychosis, 16.36% (n = 9) with mania and 9.09% (n = 5) with catatonia. There were no statistically significant changes in NSE (*p* = 0.288) and S-100b (*p =* 0.243) levels. We found no evidence that ECT induced neuronal damage based on NSE and S-100b protein levels measured in the serum of patients before and after treatment.

## 1. Introduction

Electroconvulsive therapy (ECT) is considered one of the most effective treatments for psychiatric disorders. For almost 70 years, ECT has proven effective in the treatment of depression, mania, catatonia and psychosis [1]. Several guides and expert consensus recommendations [2,3] have been developed to direct the administration of ECT; however, its mechanism of action remains elusive. Approximately 100 theories about its potential therapeutic targets have been postulated [4]. Questions about safety and adverse events have also been posed. The anticonvulsive hypothesis, as a mechanism of action of ECT, states that seizures induced during treatment cause the release of neurotransmitters in the central nervous system (CNS) that reduce brain excitability and induce seizure termination. These effects are essential to ECT’s therapeutic effect in psychiatric disorders [2,5,6,7].

As in neurological convulsive disorders, it is presumed that seizures induced during ECT administration cause toxicity and potentially neuronal and glial cell death and are responsible for the cognitive disturbances following treatment [8]. These cognitive disturbances could reflect tissue damage as a consequence of ECT even with normal imaging findings. Because the intercellular space in the brain is in direct contact with the cerebrospinal fluid (CSF), biochemical changes in brain tissue may be reflected in the CSF. Therefore, different types of brain damage may be evaluated by measuring brain-derived proteins in the CSF [9,10].

Biomarkers for cerebral damage investigated in many neurological disorders such as stroke, central nervous system (CNS) tumors and epilepsy include neuron-specific enolase (NSE), S-100b protein (S-100 b), creatine kinase, lactate dehydrogenase, glial fibrillary acidic protein and myelin basic protein; however, measurement of NSE and S-100b protein is widely preferred due to good sensitivity for neuronal tissue damage and relatively low-cost processing [11,12].

NSE is a metalloenzyme that catalyzes the conversion of 2-phosphoglycerate to phosphoenolpyruvate during glycolysis; it possesses three subunits called α, β and γ and five possible isoforms called αα, αβ, αγ, ββ and γγ. Nervous tissue expresses isoforms containing the γ subunit; αγ form is expressed in microglia, astrocytes and oligodendrocytes, while the γγ form is expressed in neurons [13,14].

S-100b protein is part of a family of low-molecular-weight proteins formed by two immunologically different subunits, which are α and β, respectively. Main isoforms of this protein include: S-100ao-αα, S-100a-αβ and S-100b-ββ. Astroglial cells express αβ and ββ isoforms which regulate many intracellular processes such as signal transduction, cellular differentiation, RNA transcription and cell cycle progression [15,16]. 

A broad range of CNS disorders, from neurodegenerative diseases to CNS tumors, can increase CSF and serum levels of NSE and S-100b. Accelerated cell proliferation and cell death cause the release of these markers to the extracellular space and blood, allowing for their measurement and utilization as a proxy for tissue damage [17,18].

Proposed mechanisms for ECT-related cerebral damage include the increased cellular energy consumption during seizure, loss of GABA receptors and toxicity mediated by glutamate release into the synaptic cleft. Glutamate binds to NMDA and AMPA receptors, resulting in early neuron apoptosis due to abnormal calcium influx [19]. 

Several studies have investigated the optimal cutoff value for NSE and S-100b to accurately diagnose neuronal cell damage, with mixed results. Despite some authors supporting the idea of a high NSE (>20 ng/mL) cutoff value to avoid false positives, current test kits have a reference interval of 10–20 ng/mL and most of them consider abnormal levels to be above 12.5 ng/mL. Evidence on S-100b levels is more straightforward; it is accepted that levels higher than 0.15 µg/L are abnormal [20,21].

The aim of this study is to investigate the influence of ECT on levels of NSE and S-100b as a proxy for neuronal cell damage in a clinical sample of patients with diagnoses of depression, mania, catatonia and psychosis during inpatient treatment. 

## 2. Methods

### 2.1. Study Participants

Our study was conducted in the Neuropsychiatry Department of the National Institute of Neurology and Neurosurgery Manuel Velasco Suárez in Mexico City, from 1 October 2009 to 1 June 2010. All participants provided written informed consent. Participants under the age of 18 provided assent in conjunction with parental consent.

This is a naturalistic observational prospective analytic study. We included 55 patients, 38 women and 17 men, aged 15–70 years with clinical indications of ECT. Patients who met the inclusion criteria were medicated with psychotropic drugs specific for their underlying condition. The clinical symptoms of patients were assessed individually using rating scales specific for their diagnosis. These scales included the Hamilton Depression Rating Scale (0–9 = no depression, 10–18 = mild depression, 19–25 = moderate depression, and 26 or more = severe depression); the Young Mania Rating Scale (an instrument for quantifying symptoms consisting of 11 items, with 5 options each, reflecting increasing degrees of symptomatic intensity; the total range of the scale is 0–60 points, with no established cutoff points or severity levels, and various criteria can be found in the literature); the Bush–Francis Catatonia Rating Scale (a diagnosis is made with 4 positive items); and PANSS, the Positive and Negative Syndrome Scale for psychosis (measuring the severity of symptoms present in schizophrenia, allowing for the evaluation of changes in response to treatment; each variable is scored according to severity from 1 = absent to 7 = extremely severe, with a cutoff point of 60). These assessments were performed before and after ECT.

Inclusion Criteria

(1)Men and women over 15 years of age.(2)Diagnosis of the following psychiatric disorders in which ECT is part of the treatment algorithm: depression (unipolar or bipolar), mania, catatonia or schizophrenia and psychosis Not Otherwise Specified (NOS), according to Diagnosis and Statistics Manual Fourth Edition (DSM-IV; APA, 1994 [22]).(3)Diagnosis of psychiatric disorders resistant to pharmacological treatment based on guidelines from the National Institute of Clinical Excellence (NICE) available at the time of the study.(4)Major indication for the use of ECT due to symptom severity in the included psychiatric disorders.

Exclusion Criteria

(1)Concomitant medical illness such as diabetes, hypertension, liver failure, renal failure, acute ischemic stroke, hemorrhagic stroke, melanoma, acute traumatic brain injury and cancer of any etiology.(2)Current diagnosis of substance use disorder.(3)Contraindication to ECT or anesthesia.

Mini-Mental State Examination (MMSE) was also conducted for cognitive monitoring, CGI (Clinical Global Impression) was conducted for illness severity, and BPRS (Brief Psychiatric Rating Scale) was conducted for the aforementioned assessments before the first ECT session and after the last ECT session.

All patients provided informed consent, and the study was approved by the bioethics committee.

ECT was performed according to the guidelines (clinical protocols) of the Neuropsychiatry Department of the National Institute of Neurology in Mexico, which utilizes a Thymatron TM DGx device from Somatics (for detailed specification and characteristics of electrical stimulus generation, see Appendix A). The decision to use ECT in the management of the patient along with the number of sessions was made by the treating physician based on their clinical experience, treatment guidelines available at that time for each underlying diagnosis and the patient’s clinical evolution.

### 2.2. Measurement of S-100b and NE

Venous blood for biomarker assays was collected using EDTA vacuum tubes using standard phlebotomy procedures before the first ECT session (baseline level) and immediately after the last ECT session. Plasma was collected after centrifugation. To guarantee consistent analyses and reduce assay variation, blood samples were centrifugated within 15 min after extraction and placed in cold storage with temperatures from +2 °C to +8 °C until processing (maximum time from blood extraction to processing was 3 h). For labeling of the samples, we used a code that incorporated date of extraction and protocol number (see Institutional Review Board Statement section). The concentrations of S-100b and NSE were measured by immunoluminometric analysis from two centers (sandwich type) using paramagnetic microparticles coated in an automatic analyzer (LIAISON; AB Sangtec Medical/DiaSorin, Bromma, Sweden; LIAISON^®^ Sangtec 100^®^).

### 2.3. Statistical Analysis 

We performed a descriptive analysis of dimensional variables with central tendency measures. Normality was assessed using the Kolmogorov–Smirnov test. The baseline scores of the scales and serum levels of S-100b and NSE were compared with the post-treatment scores and serum levels by means of a paired-sample *t* test. Statistical significance level for the variables was stablished at *p* < 0.05.

Statistical analysis was performed using SPSS Statistics 28.0 for Windows (IBM SPSS Statistics, 2021).

### 2.4. Ethical Implications of the Study

The protocol complies with the General Health Law of Mexico and the Clinical Research Regulation of the National Institute of Neurology and Neurosurgery “Manuel Velasco Suárez.” For its execution, the hormone laboratory of INNN received donations of necessary reagents from the equipment distributor since measurements of NSE and S-100b are not part of the routine blood test performed in our clinical practice. 

## 3. Results 

The analysis included 38 female patients (69.09%) and 17 male patients (30.91%) with a mean age of 38.6 years (SD 15.1) within the range of 15 to 70 years. Of the 55 patients, 52.73% (n = 29) were diagnosed with depression, 21.82% (n = 12) with schizophrenia or other psychosis, 16.36% (n = 9) with mania and 9.09% (n = 5) with catatonia. Patient characteristics are shown in Table 1.

### 3.1. Characteristics Related to Interventions 

The number of ECT sessions ranged from a minimum of 3 to a maximum of 15, with a mean of 6.4 (SD 2.4). The average stimulus intensity was 29.2 (SD 14.1), within the range of 10 to 70. The vast majority were bilateral (92.73%). The initial percentage of electrical stimulus used was calculated by dividing the patient’s age by two. 

### 3.2. Initial–Final Comparison 

The mean differences in serum levels of S-100b and NSE were 0.009 and 3.95 (*p* = 0.243; *p* = 0.288, respectively); no significant change in the serum levels of either marker was found. With the exception of the Bush–Francis Catatonia Rating Scale, all clinical assessment rating scales showed a significant reduction in their scores after ECT (*p* < 0.000; Table 2).

## 4. Discussion

Various studies have strongly supported the safety of Electroconvulsive Therapy [23,24,25,26]. However, the possibility of neuronal loss or damage in a minority of patients cannot be excluded, with the therapy being associated with cognitive adverse effects caused by neuronal damage [27]. Nevertheless, controversy continues [28,29].

In line with previous studies that analyzed serum levels of S-100b and NSE from samples taken before ECT and after repeated sessions [30,31,32], our study found no significant change in the levels of either S-100b and NSE. 

The exact half-life of S-100b has not been definitively established. Reports suggest that its half-life can range from 60 min to 24 h [33]; however, Jönsson et al. [34] propose that it is short, approximately 20–25 min. This was taken into consideration in our study, as samples were taken immediately after ECT.

Altogether, our study adds to existing information supporting the safety of ECT and reaffirms the lack of evidence on ECT-induced neuronal cell damage.

Considering brain plasticity, memory and learning in relation to S-100b [35,36], it was found that high concentrations of S-100b can negatively affect spatial learning [37]. This contrasts with reports by Angelink et al. [30], who demonstrated that patients with high post-ECT serum levels of NSE and S-100b showed better cognitive performance, perhaps due to improvement in neuronal plasticity after ECT. In our study, there was a significant increase in MMSE, suggesting improvement in patients’ general cognition.

Regarding clinimetrics performed in our study, all clinical assessment scales showed a significant reduction (*p* < 0.000), except Bush–Francis (*p* = 0.089). This speaks to the efficacy of ECT in the treatment of four main psychiatric disorders. The lack of significance in the change in the Bush–Francis score may be due to sample size.

One limitation of our study was the heterogeneity of our patients and the varying number of ECT sessions used for sample collection. Studies have reported that S-100b levels may be age-dependent, but this has not been confirmed [38].

## 5. Conclusions

Electroconvulsive therapy (ECT) is considered a safe and effective treatment for psychiatric disorders with specific indications. Cognitive adverse effects have led critics to question the safety and the possibility of neuronal damage induced by ECT. We found no evidence of ECT-induced neuronal damage based on NSE and S-100b protein levels measured in the serum of patients before and after treatment.

## Figures and Tables

**Table 1 brainsci-14-00822-t001:** Patient characteristics.

Features	N	Mean	SD	Min.	Max.
Age	55	38.6	15.1	15	70
**Sex**	**N**	**%**			
Male	17	30.91			
Female	38	69.09			
**Diagnosis**	**N**	**%**			
Depression	29	52.73			
Catatonia	5	9.09			
Mania	9	16.36			
Schizophrenia and psychosis NOS	12	21.82			
**Position of electrodes**	**N**	**%**			
Bilateral	51	92.73			
Unilateral	4	7.27			
**Number of ECT sessions**	**N**	**Mean**	**SD**	**Min.**	**Max.**
	55	6.43	2.40	3	15
**Stimulus intensity**	**N**	**Mean**	**SD**	**Min.**	**Max.**
	55	29.2	14.1	10	70

SD: Standard Deviation; NOS: Not Otherwise Specified; ECT: Electroconvulsive Therapy.

**Table 2 brainsci-14-00822-t002:** Comparative analysis of clinical assessment rating scales and serum levels of S-100b and NSE before and after ECT.

	Baseline (SD)	After ECT (SD)	Mean Difference	*p*
Marker				
S-100b	0.0831 (0.058)	0.074 (0.049)	0.009	0.243
NSE	16.87 (29.27)	12.92(11.84)	3.95	0.288
Scale				
CGI	5.05 (0.78)	2.85 (0.84)	2.2	<0.000
BPRS	23.75 (7.32)	10.71 (6.74)	13.03	<0.000
MMSE	20.49 (11.05)	27.14 (7.03)	−6.65	<0.000
HAMD	21.10 (3.50)	9.34 (4.80)	11.75	<0.000
Bush–Francis	17.20 (9.57)	5.20 (6.57)	12	0.089
Young	24.66 (6.83)	7.11 (4.42)	17.55	<0.000
PANSS	105.33 (14.19)	68.66 (17.77)	36.66	<0.000

CGI: Clinical Global Impression scale; BPRS: Brief Psychiatric Rating Scale; MMSE: Mini-Mental State Examination; HAMD: Hamilton Depression Rating Scale; Bush–Francis: Bush–Francis Catatonia Rating Scale; Young: Young Mania Rating Scale; PANSS: Positive and Negative Syndrome Scale for schizophrenia; S-100b: S-100b protein; NSE: neuron-specific enolase.

## Data Availability

The original contributions presented in the study are included in the article/Appendix A; further inquiries can be directed to the corresponding author.

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
