# Peer review of "Neurological Damage Measured by S-100b and Neuron-Specific Enolase in Patients Treated with Electroconvulsive Therapy"

_brainsci, 2024, doi:10.3390/brainsci14080822_

Round 1

Reviewer 1 Report

Comments and Suggestions for Authors

Introduction

The introduction of the article provides an overview of Electroconvulsive Therapy (ECT) as a treatment for psychiatric disorders. The historical context and efficacy of ECT are well articulated, with references to the treatment of depression, mania, catatonia, and psychosis. The introduction also highlights ECT's elusive mechanism of action and the controversy surrounding its safety, particularly regarding potential neuronal and glial cell damage. Discussing biomarkers such as NSE and S-100B proteins is relevant, given their roles in evaluating neuronal tissue damage.

Critical Remarks:

  • The choice of tested proteins should be better justified. The term "most commonly used" is not sufficient.
  • An interesting supplement to the introduction would be a graphical presentation of the mechanism of action of the S100B and NSE proteins selected for analysis.
  • The purpose of the study should be supplemented with the disease entities that were included in the study.

Methods

The methods section describes the study design, participants, inclusion and exclusion criteria, assessments, and statistical analyses. The study included 55 patients with a range of psychiatric disorders who underwent ECT. Cognitive monitoring and various clinical assessments were conducted before and after ECT sessions. Serum levels of S-100B and NSE were measured using immunoluminometric analysis.

Critical Remarks:

  • The description of the study group is very poor and insufficient. It should be supplemented with all available data, e.g., demographics, results of basic laboratory tests, medical history, etc.
  • The diagnosis confirmation process and the diagnostic tools/criteria should be detailed.
  • The description needs a detailed account of patients' qualifications for the procedure, especially since it is reserved mainly for drug-resistant patients. Information should be provided on what national or international recommendations were followed in this regard. The inclusion and exclusion criteria in their current form are unacceptable.
  • A detailed description of the banking and storage of biological material needs to be provided.
  • The description of protein measurement using a technique that requires a set of reagents and a protocol should be completed.
  • The justification for the selection of patients participating in the study, ranging from depression and mania to schizophrenia, was omitted.
  • The very small number of patients in each group indicates poor study planning. There was no assessment of the number of patients needed to recruit to achieve at least 80% study power.
  • The statistical analysis should be separated into a distinct paragraph. The normal distribution of individual variables was not checked, and further selection of the tests used was not justified. The power of the study was not checked, and appropriate statistical testing corrections were not applied.
  • The influence of gender on the analyzed variables (confounding variable) should be examined. If there is no relationship, omit the division by gender due to the small sample sizes, e.g., female/catatonia N=5.
  • Lack of information regarding obtaining the consent of the local bioethics committee for the research, information provided to the patient and the formula for giving their informed consent

Results

The results section presents a statistical analysis of the changes in serum S-100b and NSE levels before and after ECT. The data showed a decrease in both biomarkers, with S-100b decreasing by 10.0% and NSE by 23.5%, though these changes were not statistically significant. The clinical scales (CGI, BPRS, MMSE, Hamilton, Young, and PANSS) showed significant improvements, indicating the clinical efficacy of ECT.

Critical Remarks:

  • The results section contains a lot of numerical data that could be better visualized with graphs or charts.
  • The lack of significant change in biomarkers might need more discussion on the potential reasons and implications.
  • No correlation analyses were performed with the results of all the tests and questionnaires used.
  • Re-analysis should be performed in accordance with the revised description in the "statistical analysis" paragraph.

Discussion

The discussion interprets the findings in the context of existing literature, highlighting the safety of ECT and the lack of significant neuronal damage as indicated by stable NSE and S-100B levels. It also discusses the gender differences observed in S-100B levels and correlates clinical improvements with biomarker levels.

Critical Remarks:

  • The discussion does not allow for reliable conclusions to be drawn based on the results presented to very small groups of patients and incorrect methodology. Considering the clinical improvements, the discussion could benefit from a more in-depth exploration of why there was no significant change in biomarkers.
  • More emphasis on the study's limitations, such as sample size and heterogeneity, would provide a balanced perspective.

Conclusion

The conclusion summarises the study's findings, reiterating the safety and efficacy of ECT without evidence of neuronal damage based on serum biomarkers. It emphasizes the clinical improvements observed in patients.

Critical Remarks:

  • The conclusions should be revised based on a re-analysis of the data
  • The conclusion could be strengthened by suggesting further research directions to explore the relationship between ECT and neuronal biomarkers.

Overall Evaluation

The article examines the effects of ECT on serum biomarkers NSE and S-100B. The methodology is poorly described and indicates the authors' lack of a uniform and reliable study design. The results require re-analysis in accordance with the accepted principles of scientific publications. The discussion must be updated with reliable results and supplemented with any methodological limitations affecting the assessment of scientific credibility. The presentation could be improved for readability and visual engagement. Addressing the limitations more thoroughly and suggesting future research directions further enhance the article's contribution to the field. The manuscript in its current form does not provide new or extended information in the presented field and is of a replication nature.

Author Response

Comment 1

  • The choice of tested proteins should be better justified. The term "most commonly used" is not sufficient.
  • The purpose of the study should be supplemented with the disease entities that were included in the study.

Reponse 1

  • Line 47-49, we state good sensibility for neuronal damage of NSE and S-100b levels with a relatively low cost as th reason to use this markers.
  • Agree, we expanded the purpose description

Comment 2 

  • The description of the study group is very poor and insufficient. It should be supplemented with all available data, e.g., demographics, results of basic laboratory tests, medical history, etc.
  • The diagnosis confirmation process and the diagnostic tools/criteria should be detailed.
  • The description needs a detailed account of patients' qualifications for the procedure, especially since it is reserved mainly for drug-resistant patients. Information should be provided on what national or international recommendations were followed in this regard. The inclusion and exclusion criteria in their current form are unacceptable.
  • A detailed description of the banking and storage of biological material needs to be provided.
  • The description of protein measurement using a technique that requires a set of reagents and a protocol should be completed.
  • The justification for the selection of patients participating in the study, ranging from depression and mania to schizophrenia, was omitted.
  • The very small number of patients in each group indicates poor study planning. There was no assessment of the number of patients needed to recruit to achieve at least 80% study power.
  • The statistical analysis should be separated into a distinct paragraph. The normal distribution of individual variables was not checked, and further selection of the tests used was not justified. The power of the study was not checked, and appropriate statistical testing corrections were not applied.
  • The influence of gender on the analyzed variables (confounding variable) should be examined. If there is no relationship, omit the division by gender due to the small sample sizes, e.g., female/catatonia N=5.
  • Lack of information regarding obtaining the consent of the local bioethics committee for the research, information provided to the patient and the formula for giving their informed consent

Response 2

  • We expanded description for diagnostic and refractoriness to pharmacological treatment criteria mentioning the use of DSM IV for diagnosis and NICE guidelines available at the time of the study for refractoriness
  • As this was a naturalistic study, the decision to include a patient was based solely on the indication to use ECT as part of the treatment; as this therapy is limited to the aforementioned psychiatric disorders in the inclusion criteria section
  • Regarding statistical analysis we corrected the procedure: we tested normality using Kolmogorov-Smirnov and then simplified our previous work by only performing paired t test analysis of the variables.

Discussion and conclussion were reworked according to reanalysis.

Reviewer 2 Report

Comments and Suggestions for Authors

The topic of the manuscript is actual and interesting, but I still have serious concerns. 

The methodology is insufficient, the range of applied ECT sessions is wide and unclear, the authors failed to define the number, frequency, and the sites of electrodes application. It should be stated in the MM section. In my opinion, the results are affected by different categories of patients included in the study (different diagnosis). Also, the authors failed to respond if the analyses of the indicators S100B and NSE are part of the standard clinical practice, and the results from 2007 to 2010 retrospectively analyzed and presented in this manuscript. It seems confused when the analyses of the mentioned markers were done. 

The lack of methodology impact on the reliability of results. Also, the presented results have to be uniform considering SD or SEM. 

Author Response

Comment 1

The methodology is insufficient, the range of applied ECT sessions is wide and unclear, the authors failed to define the number, frequency, and the sites of electrodes application. It should be stated in the MM section. In my opinion, the results are affected by different categories of patients included in the study (different diagnosis). Also, the authors failed to respond if the analyses of the indicators S100B and NSE are part of the standard clinical practice, and the results from 2007 to 2010 retrospectively analyzed and presented in this manuscript. It seems confused when the analyses of the mentioned markers were done. 

The lack of methodology impact on the reliability of results. Also, the presented results have to be uniform considering SD or SEM. 

Reponse 

Thank you for your comment. 

We provided available data about electrode positioning. 

We remade the statistical analysis to comply with the scientific method.

We expanded the description fo the methods regarding blood sample processing, as they were processed within the following 3 hours after obtention. 

Line 150-152: we state that measurement of NSE and S-100b are not part of routinary blood test performed in our clinical practice. 
